# Selecting 16S rRNA Primers for Microbiome Analysis in a Host–Microbe System: The Case of the Jellyfish *Rhopilema nomadica*

**DOI:** 10.3390/microorganisms11040955

**Published:** 2023-04-06

**Authors:** Noga Barak, Eduard Fadeev, Vera Brekhman, Dikla Aharonovich, Tamar Lotan, Daniel Sher

**Affiliations:** 1Marine Biology Department, The Leon H. Charney School of Marine Sciences, University of Haifa, Haifa 3498838, Israel; nbarak13@campus.haifa.ac.il (N.B.);; 2Department of Functional and Evolutionary Ecology, University of Vienna, 1030 Vienna, Austria

**Keywords:** cnidaria, jellyfish, amplicon sequencing, next-generation sequencing, microbial community, method comparison, universal primers

## Abstract

Amplicon sequencing of the 16S rRNA gene is extensively used to characterize bacterial communities, including those living in association with eukaryotic hosts. Deciding which region of the 16S rRNA gene to analyze and selecting the appropriate PCR primers remains a major decision when initiating any new microbiome study. Based on a detailed literature survey of studies focusing on cnidarian microbiomes, we compared three commonly used primers targeting different hypervariable regions of the 16S rRNA gene, V1V2, V3V4, and V4V5, using the jellyfish *Rhopilema nomadica* as a model. Although all primers exhibit a similar pattern in bacterial community composition, the performance of the V3V4 primer set was superior to V1V2 and V4V5. The V1V2 primers misclassified bacteria from the Bacilli class and exhibited low classification resolution for Rickettsiales, which represent the second most abundant 16S rRNA gene sequence in all the primers. The V4V5 primer set detected almost the same community composition as the V3V4, but the ability of these primers to also amplify the eukaryotic 18S rRNA gene may hinder bacterial community observations. However, after overcoming the challenges possessed by each one of those primers, we found that all three of them show very similar bacterial community dynamics and compositions. Nevertheless, based on our results, we propose that the V3V4 primer set is potentially the most suitable for studying jellyfish-associated bacterial communities. Our results suggest that, at least for jellyfish samples, it may be feasible to directly compare microbial community estimates from different studies, each using different primers but otherwise similar experimental protocols. More generally, we recommend specifically testing different primers for each new organism or system as a prelude to large-scale 16S rRNA gene amplicon analyses, especially of previously unstudied host–microbe associations.

## 1. Introduction

The existence of prokaryotic microbes dates long before the first eukaryotic animals appeared and it is likely that host–bacteria symbioses have existed for hundreds of millions of years [1,2]. These symbioses are not only important and can influence health and disease of many hosts, but they can also help us understand evolution and development processes [3,4]. In recent years, alongside studies of bacteria as pathogens, there has been increasing evidence on bacteria playing significant roles in shaping the phenotype, development, behavior, and fitness of various hosts [5,6,7,8]. The host-associated microbial community can provide numerous beneficial functions, including nutrition, assisting in the maturation of the immune system or protection from pathogenic infections [9,10]. These important roles played by symbiotic bacteria in the lives of extant eukaryotic hosts raise the question of how these symbioses may have contributed to host evolution, as postulated by the hologenome theory of evolution [11,12]. Answering such questions could be facilitated by the study of ancient phyla such as cnidarians and their associated symbionts.

Cnidarians, which include sea anemones, corals, hydrozoans, jellyfish, and parasitic myxozoas, are one of the oldest animal phyla, having evolved 700 million years ago [13]. Microbes associated with corals, hydra, jellyfish, and other cnidarians have been extensively investigated since they impact the health and fitness of the cnidarian host [2,3,9,14,15,16]. These studies suggest that the cnidarian microbiome differs between closely related organisms, is distinct from that of the surrounding water, and varies as the life cycle of the host progresses, e.g., from benthic to pelagic [17]. Several studies have also shown differences in microbial community structure between healthy and diseased cnidarian tissue, thus linking the health of the host [9,18]. Given the importance of cnidarians (especially corals and some hydrozoa) as reef-building organisms, and the prevalence of jellyfish blooms worldwide, understanding how the cnidarian microbiome is related to host health is important, e.g., for conservation or mitigation efforts [17,19,20]. However, research on the structure of cnidarian-associated bacteria is challenging [21] and is currently conducted using highly diverse methodologies. The technical discrepancies between the studies impede the establishment of a general overview on the cnidarian microbiome.

The most commonly used methodology for taxonomic profiling of bacterial communities is amplification and sequencing of (part of) the 16S rRNA gene. Since the 16S rRNA gene consists of nine variable regions which can be targeted by different primers (in different combinations) for sequencing, choosing the right primer combination is important (Figure 1A). For instance, The Earth Microbiome Project, which aims to characterize and interpret the microbial diversity and functional potential of thousands of environmental samples, uses primers that target the V4 hypervariable region [22]. In contrast, the Human Microbiome Project uses mostly primers spanning the V1V3 and V3V5 regions [23], some studies focusing on cross-taxa comparisons, clinical studies, and the gut microbiome employ primers spanning the V3V4 and V1V2 regions [24,25,26], and many studies of the ocean target the V4V5 region [27]. Even within a more limited range of environments or hosts, such as in studies of cnidarian microbiomes, there is no standard primer set used in all studies. Since 2015, 152 studies have investigated the taxonomic composition of Cnidaria-associated bacteria, using 11 different primer sets, each targeting different regions of the 16S rRNA gene (Figure 1B). Most of the studies focused on corals, and most targeted the V4 or V3V4 regions, yet many studies of jellyfish-associated bacterial communities targeted the V1V2 region. Therefore, the question of whether there are “optimal” primer sets for different environments or hosts, including cnidarians, remains open [28,29,30,31,32].

Our model jellyfish, *Rhopilema nomadica*, is known to form massive blooms that have significant ecological and economic impacts [33,33,34]. These blooms are characterized by a rapid appearance and disappearance, and much remains unknown about the complex interplay between the blooms and other factors in the marine system, including the role of the jellyfish’s symbiotic microbial community. We have therefore initiated a large, multi-annual study of the *R. nomadica* microbiome in the Eastern Mediterranean, aiming to identify changes in the jellyfish microbiome that may be associated with bloom initiation and collapse (e.g., potential pathogens). Here, we describe the first stage of this project, namely, a comparison of three commonly used 16S rRNA primer sets (V1V2, V3V4, and V4V5) to select the most appropriate primers for this study (Figure 1A). We examined the performance of the different primer sets in terms of the number of useful sequences, assessments of diversity, and the taxonomical assignment as individual ASVs (Amplicon Sequence Variants).

Furthermore, we asked whether it is possible to combine the results of the different primer sets. While our analysis was focused on a jellyfish, Rhopilema nomadica, we believe the systematic comparison between different 16S rRNA primers can provide useful insights for the study of host-associated bacteria in general. Amplicon sequencing of the 16S rRNA gene is extensively used to characterize bacterial communities, including those living in association with eukaryotic hosts. Deciding which region of the 16S rRNA gene to analyze and selecting the appropriate PCR primers remains a major decision when initiating any new microbiome study. Based on a detailed literature survey of studies focusing on cnidarian microbiomes, we compared three commonly used primers targeting different hypervariable regions of the 16S rRNA gene, V1V2, V3V4, and V4V5, using the jellyfish Rhopilema nomadica as a model. Although all primers exhibit a similar pattern in bacterial community composition, the performance of the V3V4 primer set was superior to V1V2 and V4V5. The V1V2 primers misclassified bacteria from the Bacilli class and exhibited low classification resolution for Rickettsiales, which represent the second most abundant 16S rRNA gene sequence in all the primers. The V4V5 primer set detected almost the same community composition as the V3V4, but the ability of these primers to also amplify the eukaryotic 18S rRNA gene may hinder bacterial community observations. However, after overcoming the challenges possessed by each one of those primers, we found that all three of them show very similar bacterial community dynamics and compositions. Nevertheless, based on our results, we propose that the V3V4 primer set is potentially the most suitable for studying jellyfish-associated bacterial communities. Our results suggest that, at least for jellyfish samples, it may be feasible to directly compare microbial community estimates from different studies, each using different primers but otherwise similar experimental protocols. More generally, we recommend specifically testing different primers for each new organism or system as a prelude to large-scale 16S rRNA gene amplicon analyses, especially of previously unstudied host–microbe associations.

## 2. Materials and Methods

### 2.1. Jellyfish and Sea Water Samples Collection and Preparation

On the 19th of July 2020, during a major jellyfish bloom in the eastern Mediterranean, three female *Rhopilema nomadica* jellyfish of similar sizes (24–32 cm) were collected from Haifa Bay, Israel (32°50′22.9″ N 35°00′04.8″ E). The jellyfish were caught individually by swimmers using round buckets to avoid damage to the animals and then gently transferred to 80 L containers on the boat filled with ambient seawater. All the collected jellyfish were active and healthy. Animal dissection and sample preservation were performed less than two hours after specimen collection. The diameter of each jellyfish was measured, and different tissues were collected from each jellyfish—bell, tentacles, gastrovascular canals, gonads, and rhopalium—using sterile tool kits. After the collection of each tissue, the dissection tools went through a process of cleaning using 1% sodium hypochlorite, DNA AWAY™ (Thermo Fisher, Waltham, MA, USA), 70% Ethanol, and finally, Ultrapure (miliQ) Water. Triplicates from each tissue were collected and placed immediately on dry ice. Upon arrival at the lab, the frozen tubes were kept in −80 °C for later use. To identify the gender of the jellyfish, gonad samples from each jellyfish were maintained on ice and observed using a dissecting microscope (Zeiss Axio Imager M2). In addition to the jellyfish samples, five liters of seawater was filtered on a Sterivex filter cartridge (0.22 μm), 1 mL of preservation/lysis solution was added (40 mM EDTA, 50 mM Tris, pH 8.3, 0.75 M Sucrose), and the samples were frozen as described above.

Bell, gastrovascular canals, gonads, and rhopalium samples were homogenized using a bead beater (TissueLyser II Qiagen, Hilden, Germany) followed by 1 h treatment with lysozyme at 37 °C (Merck, Darmstadt, Germany 100 mg/mL) and 1 h proteinase k at 55 °C (Promrga, Madison, WI, USA, 20 mg/mL). DNA of the total microbial community together with jellyfish DNA was extracted with the ZymoBIOMICS DNA Miniprep Kit (Zymo Research, Irvine, CA, USA) following the manufacturer’s protocol. The Sterivex filter (Millipore, Burlington, MA, USA) was cut as described by Cruaud et al., 2017 [35], and the filter was placed in the ZR BashingBead™ Lysis Tubes from the ZymoBIOMICS DNA Miniprep Kit. Subsequent DNA extraction was performed as described above for the tissue samples.

For standardization, a commercially available mock microbial community standard was used (ZymoBIOMICS™, Zymo Research). The mock community was extracted using 75 µL per prep as recommended by the manufacturer following the same protocol as all other samples.

### 2.2. 16S rRNA Gene Library Preparation and Sequencing

Three different hypervariable regions of the bacterial 16S rRNA gene were amplified using aliquots of the isolated DNA from each sample—V1V2, V3V4, and V4V5 (Appendix A). Amplicons were generated using a two-stage polymerase chain reaction (PCR) amplification protocol as described previously [36]. The primers contained 5′ common sequence tags (known as common sequence 1 and 2, CS1 and CS2). First stage PCR amplification was carried out using the DreamTaq Green PCR master mix (M/s Thermo scientific, Waltham, USA). Briefly, each 50 µL reaction mix contained 25 µL of DreamTaq Green PCR master mix (2X), 1 µL (10 µM) each of forward and reverse primers, 2 µL of template DNA, and 21 µL of nuclease-free water. We note that this enzyme is not a proofreading polymerase, which may be more appropriate when ASV-level dynamics are expected. The amplification parameters were set as follows: 95 °C for 5 min, followed by 32 cycles at 95 °C for 30 s, 52 °C for 45 s, and 72 °C for 1 min, and a final extension phase of 72 °C for 7 min. Products were verified on a 1% agarose gel before moving forward to the 2nd stage. V1V2 and V3V4 produced single band while V4V5 produced different molecular mass bands (see detailed discussion below). One microliter of PCR product from the first stage amplification was used as a template for the 2nd stage, without cleanup. Cycling conditions were 98 °C for 2 min, followed by 8 cycles of 98 °C for 10 s, 60 °C for 1 min, and 68 °C for 1 min. Libraries were then pooled and sequenced with a 15% phiX spike-in on an Illumina MiSeq sequencer employing V3 chemistry (2 × 300 base paired-end reads). Library preparation and sequencing were performed at the Genomics and Microbiome Core Facility (GMCF; Rush University, IL, USA).

### 2.3. Literature Survey of Microbiome Studies in Cnidaria

Publications were searched on PubMed using the following query: (16S rRNA) AND (“bacterial composition” OR microbiome OR microbiota) AND (Cnidaria OR jellyfish OR Sea Anemone OR hydra OR hydrozoa OR coral OR myxozoa). Only publications between the years 2015–2022 were selected. The query resulted in 232 papers where 138 were selected after filtration. The papers we filtered out included bacteria isolation reports or studies on non-Cnidaria hosts that were associated with coral reefs. A complete list of papers and details on each specific term are in the Appendix A.

### 2.4. Data Analysis and Visualization

Quality control of the raw paired-end reads was performed before the analysis using the FastQC v0.11.9 tool. Further analysis was conducted using R 4.1.0 in RStudio v1.4.1717-3. All of the amplicons libraries were processes using DADA2 [37] and following the recommended workflow [38]. The forward reads were trimmed to 220 bp (V1V2), 240 bp (V3V4), or 260 bp (V4V5) and the reverse reads were trimmed to 200 bp (V3V4) or 215 bp (V3V4 and V1V2). Parameters for trimming were set based on FastQC reads quality results which varied slightly between primers. We also trimmed primer sequences so these would not affect any downstream analysis. Forward and reverse reads were merged based on a minimum overlap of 8 bp, chimeras were filtered out, and an amplicon sequence variants (ASVs) table was created. The representative ASVs were taxonomically classified against the SILVA 16S rRNA gene reference database v138.1 [39]. ASVs that were not classified at the phylum level or were not assigned to bacterial lineages (including those assigned to mitochondria and chloroplasts) were excluded from further analysis.

Management of the data (organization of tables) was performed using the R packages “phyloseq” v1.40.0 [40] and dplyr v1.0.9 [41]. Plots were generated using the R package “ggplot2” v3.3.6 [42]. Alpha diversity indexes were calculated using the function “estimate_richness” in “phyloseq”. The Kruskal–Wallis rank sum test was conducted using the R package “rstatix” v0.7.0 [43]. Non-metric multidimensional scaling plots were created to determine differences in the bacteria communities of different primer sets using R packages “phyloseq” v1.40.0 [40] and “vegan” v2.6.2 [38]. For the NMDS, Bray–Curtis distance between samples was calculated based for relative abundance values of sequences (ASVs) and genus levels. Two-way permutation multivariate analysis of variance (“Two way PERMANOVA”) was conducted using the R package “vegan” v2.6.2 [44]. Venn diagrams were created using “VennDiagram” v1.7.3 [45], respectively.

Sequences of different molecular mass bands obtained with V4V5 primers were uploaded and processed via SILVAngs v.1.9.8/1.4.9 (https://www.arb-silva.de/ngs/ accessed on 4 August 2021). Visualization of the relative abundances of the different molecular mass bands was preformed using the Krona RSF display tool [46]. In some cases, relevant ASV sequences (Appendix A) were extracted and classified using SINA against the SILVA database and BLAST against the NCBI databases [47,48]. The analyses of the mock community were performed using all 16S copy sequences taken from the product info of the ZymoBIOMICS microbial community standards.

## 3. Results

### 3.1. Multiple Amplicon Sequence Lengths Affect Community Coverage

PCR amplification of the 16S rRNA gene was expected to result in a single band (~355 bp, ~510 bp, and ~450 bp for the V1V2, V3V4, and V4V5 primer sets, respectively). Indeed, the PCR results of V1V2 and V3V4 produced single bands of the expected size (~355 bp and ~510 bp, respectively). In contrast, a double band was observed in most of the samples using the V4V5 primers (~450 bp and ~600 bp, 10/15 samples, one example shown in Figure 2A). The V4V5 primer set was previously shown to also amplify the 18S gene of eukaryotes [27,49,50] (see discussion below), and indeed sequencing of DNA from each band extracted from the gel separately shows that the high molecular mass band corresponded to the amplified jellyfish 18S rRNA gene, and the low molecular mass amplicon corresponded to the amplified 16S rRNA from the prokaryotic microbial community (Figure 2B). The ratio between the intensities of the two PCR bands in the V4V5 amplicons varied between samples and jellyfish tissues without any clear pattern. Due to the length of the V4V5 18S eukaryotic amplicon, the merging procedure for paired reads in DADA2 was largely unsuccessful, resulting in far fewer usable sequences for this primer set compared with the others (Figure 2C).

### 3.2. Heterogeneity in Amplicon Length and ASVs Number when Examining the Mock Community Control

In addition to the two clearly discernible amplicons in the V4V5 primers, there was also heterogeneity in the amplicon length (after primer trimming and merging of paired ends) which differed between the primer sets. To explore this more carefully, we analyzed the results of the mock community, which contains cells from eight organisms. All of the mock community members were identified by each of the primer sets, with the V1V2 primers resulting in the closest similarity to the expected community composition (Appendix A). The V1V2 primers produced seven different amplicon lengths compared with only two for the other primer sets (Appendix A). Additionally, for 5/8 organisms, the V1V2 primer set resulted in the identification of more ASVs than expected based on the sequences of their 16S genes. (Many organisms in the mock community have multiple 16S operons. Some of these have natural sequence variations within the V1V2 region which would result in multiple different amplicon sequences, as shown in the Appendix A.) Potentially spurious ASVs were observed only for two organisms in the V3V4 amplicons, and for none in the V4V5 ones. The PCR amplifications were performed in a randomized manner where all primer sets went through the PCR stages together, and thus the differences between the V1V2 and the other primer sets are likely not due to a technical bias in the PCR amplification, nor are they likely due to the lack of proofreading in the polymerase enzyme use. In addition, the ASVs obtained in the two replicates (for each primer set) were identical and had similar relative abundance patterns (Appendix A), further supporting the conclusion that the “unexpected” ASVs are not due to polymerase error, and may represent previously undetected diversity in the mock community.

### 3.3. Similar Pattern of Diversity and Community Structure Revealed by All Three Primer Sets

All the primer sets revealed similar patterns in diversity. The total ASV number and the Simpson index showed no significant differences between the primer sets (Figure 3, ANOVA, *p*-value > 0.05). In all primers, the highest diversity was observed in the seawater and the bell tissues compared with the other tissues. In each of the primer sets, the seawater samples clustered separately from the jellyfish samples (Figure 4A), and within the jellyfish samples, the bells clustered apart from all other tissues. The grouping of the samples corresponded to the tissue type, rather than to the individual jellyfish (Figure 4A, PERMANOVA test; differences between tissue type *p*-value < 0.001 for all primers, differences between individual jellyfish *p*-value > 0.2, Appendix A).

As the regions of the 16S rRNA gene amplified by each primer set do not overlap, the resulting ASVs cannot be compared directly (e.g., aligned together). Therefore, to determine whether there are major differences between the primer sets in the observed microbial community, following Fadeev et al., (2021), we compared the datasets from the three primers based on the sequence counts at the genus rank (i.e., the lowest shared taxonomic rank). Sequences that were unclassified at the genus rank were defined at the lowest classified rank (i.e., phylum, class, order, or family). Overall, the ASVs were merged into 551 different genera and another 167 lineages that could not be classified at the genus level. The three primer sets shared 134 of the total 551 genera, with 43–133 genera unique to each primer set (Figure 5A). The shared 134 genera represent the vast majority of the 16S sequences in each dataset (>94%), and thus, the observed microbial communities (Figure 5B). The dissimilarly patterns of the combined dataset remained similar to the ASV-based datasets of individual primer sets (Figure 4B). This differences between the tissue types were observed primarily along NMDS1, whereas a clear portioning could be seen along NMDS2 between the V1V2 primers (excluding the bell tissue) and the two other primer sets (“mirror image”, Figure 4B). The PERMANOVA test showed there is a significant difference (*p*-value < 0.001) of both primer and tissue types. However, differences between tissues were correlated with 60% of the variability whereas the different primers represented 13% (PERMANOVA test; R2 = 0.6 and 0.13, respectively, Figure 4B, Appendix A). Subsequent PERMANOVA tests were conducted to compare pairwise combinations of all primers. The results of these tests showed that there were significant differences in tissue type across all primer combinations. Furthermore, significant differences were found between the V1V2 and V3V4 or V4V5 (*p*-value < 0.001), and not between V3V4 and V4V5 (*p*-value < 0.081, although this may be due to the relatively small sample size). In order to understand whether there are specific taxa responsible for the main differences between primer or tissue types, we performed a SIMPER test for both primers and tissue types (Appendix A). The results showed that two taxa, unclassified alpha proteobacteria and Cuneatibacter (Lachnospiraceae family), significantly contributed the main differences between primers (SIMPER test; ~18%, followed by the Kruskal–Wallace test; *p*-value < 0.0001). However, the same taxa were not significant when examining different tissue types (Kruskal-Wallace test *p*-value > 0.5). Therefore, while there are differences between the primer sets, to a large extent these do not mask the biological variability between the samples.

### 3.4. Discrepancies in Taxonomic Classifications between the Primers Affect Community Coverage

We next determined whether there were specific lineages identified differently by each primer set, which could explain the different ordination of the V1V2 primer amplicons in Figure 4B. One of the most abundant shared lineages (21–23% of the sequences, 11/15 samples) could be classified by the V3V4 and V4V5 primers to the order level, whereas it could only be classified to the phylum level by the V1V2 primer set (Rickettsiales vs. Alphaproteobacteria, dark and light blue, respectively, in Figure 5C). This lineage can be seen in almost all the tissue types and especially in the gonads where it is the most dominant lineage. This dominant can explain the prominent groping if the gonads in different primers compared with other tissues.

In addition to differences in the taxonomic resolution of ASV classification, we also encountered a case where the same organism was classified by the different primers as belonging to two different classes, resulting in incorrect identification of a “primer-specific” lineage. Specifically, the fraction of unique sequences was much higher for the V1V2 primer set compared with the two others (ca. 6% compared with less than 1% of all sequences in each dataset, respectively; see Figure 5B). Most of the unique sequences in the V1V2 dataset (4.5%) corresponded to the 4th most abundant sequence on the dataset and were classified as “unclassified genus” of Lachnospiraceae (yellow in Figure 5C, observed in 5/15 samples). In both the V3V4 and V4V5 primer datasets, the third most abundant sequences represent a similar fraction of the community (4–4.6%) and were classified as “unclassified Bacilli” (brown in Figure 5C). To clarify whether Lachnospiraceae and “unclassified Bacilli” reads originate from the same organisms, we extracted the relevant ASV sequences from all primer sets and classified them against both SILVA and NCBI databases using BLAST. The manual BLAST with the V1V2 and V4V5 ASV sequences resulted in classification to the genus level as Spiroplasma (Firmicutes-Bacilli-Entomoplasmatales-Spiroplasmataceae-Spiroplasma), whereas the V3V4 classification was only to the order level (Entomoplasmatales). This was in agreement with the previous results for V3V4 and V4V5 (as part of the Bacilli class), and in contrast with the V1V2 classification as Lachnospiraceae. We note that the identity level to the best hit in the SILVA database was only 82.7–87.8%. However, BLAST against the NCBI database resulted in a much higher identity level, and all the sequences were identified as Spiroplasma (99.5–100%, Appendix A).

## 4. Discussion

The sequencing of short PCR amplicons from the 16S rRNA gene (or its RNA product) is an efficient and cost-effective way to characterize the composition of bacterial communities across niches, including environmental and host-associated communities. Despite the high popularity of 16S rRNA amplicon sequencing, choosing the appropriate region of the 16S rRNA gene for sequencing is not trivial and the choice should be carried out thoughtfully. In some environments, detailed benchmarking studies compared different primer sets [30,51,52]. However, such studies are lacking in cnidarians, and are rare in marine host-associated bacteria in general. Our review of the literature discussing cnidarian microbiomes (2015–2022) shows that most studies used primer sets spanning the V4 or V3V4 regions, and most focused on corals (Figure 1B). Jellyfish-related studies tended to use the V1V2 primers, and many studies of oligotrophic seas (such as the Eastern Mediterranean) use the V4V5 primers [2,53,54]. Hence, we investigated the performance of three frequently used primer sets, which target the V1V2, V3V4, and V4V5 regions of the 16S rRNA gene, in representing the taxonomic composition of bacterial communities associated with the jellyfish *R. nomadica*. Based on our results, we selected the V3V4 primer set as the most suitable choice for our long-term studies on *R. nomadica*. Below, we discuss the identified performance characteristics of each primer set, focusing on *R. nomadica* but extending, when possible, to other organisms or ecosystems. We then ask to what extent data from multiple primer sets are comparable.

### 4.1. Amplification of Eukaryotic 18S rRNA Gene by the V4V5 Primer Set may Affect Bacterial Diversity Observations

The V4V5 primer set is extensively used in research on pelagic microbial communities [27,31,55]. This primer set was initially designed to appropriately amplify the 16S rRNA gene of the highly abundant SAR11 clade [27,56]. Further improvements of the primer set also allowed its use for the investigation of eukaryotic microbial communities, through amplification of the 18 rRNA gene [27]. We therefore tested them in this study, hoping they would facilitate future comparisons between the jellyfish and oligotrophic seawater samples, for example, to identify to what extent bacteria from the surrounding seawater colonize the jellyfish. However, samples amplified with the V4V5 primer set produced a “double band”: two differently sized amplicons corresponding to bacterial 16S and eukaryotic 18S rRNA genes. The ability of the V4V5 primer to simultaneously amplify bacteria, eukaryote, and archaea has been described before, and has led to their use as “three-domain primers” [27,49,57,58]. Recently, Yeh and co-authors [50] developed bacterial 16S and eukaryotic 18S rRNA mock communities and proposed a workflow designed for concomitant analysis of both in which the 18S amplicon data are analyzed without merging the paired ends. However, in our case the strong but non-systematic amplification of eukaryotic jellyfish host DNA caused a major loss of data during the initial stages of analysis (i.e., far fewer assembled bacterial sequences in Figure 2B, up to an 99% loss). We also speculate that such uneven amplification of host DNA may introduce biases between samples, although our dataset was not large enough to test this possibility. Therefore, based on our results, we chose not to use these primers for our large-scale jellyfish study. We recommend that, if such primers are considered for use in analyzing host-associated microbiomes (e.g., to also study micro-eukaryotic components of the microbiome), their specificity and efficiency should be tested in small scale prior to full-scale analysis.

### 4.2. Sequences of the V1V2 Region May Lead to an Inaccurate Taxonomic Classification

We chose to test the V1V2 primers in our study as they are often used in analyses of the human gut microbiome [59,60,61,62], and have also been used in microbiome studies of different Cnidaria, such as corals, hydrozoa, sea anemones, and jellyfish (Figure 1B). In our analysis, we observed three technical issues that may have biological implications. Firstly, the size distribution of the V1V2 PCR amplicons was larger than that of the other primer sets. An amplicon length variation of V1V2 and bimodal size distribution for the V3V4 16S regions were reported in the past (and was seen in our data as well); however, the data are limited [63]. The large size variation should be considered when using these primers, to make sure that the bioinformatics pipeline does not remove biologically relevant community members (some pipelines call for a stringent cutoff on the range of amplicon sequences accepted). Secondly, the V1V2 primers classified the Rickettsiales to a lower phylogenetic resolution in comparison to the other sets of primers. The Rickettsiales are a group of obligate intracellular bacteria that can be parasitic, symbiotic, or commensal with a diverse host range [64,65]. Rickettsiales are known to be pathogenic to many animals including humans and can also be found in cnidarians, including corals (*Gorgonia ventalina*, *Orbicella annularis*, and *Orbicella faveolate*), Hydrozoa (*Hydra oligactis*), and other jellyfish (*Cyanea capillata*) [66,67]. The Rickettsiales were detected in a high relative sequence abundance in most of the tissues and especially in the gonads (where they were ~60–80% of sequences) and therefore could potentially be an important member of the jellyfish microbiome. While we did not explore this observation systematically, the lower phylogenetic resolution for this clade may limit the usefulness of this primer set in samples where Rickettsiales are abundant.

Based on manual curation, using SILVA and NCBI databases, we found that abundant sequences that were taxonomically classified as Lachnospiraceae (class Clostridia) are in fact likely affiliated to Spiroplasma (class Mollicutes, see below). This was the main reason the V1V2 amplicon sequences differed from the other two primer sets in the NMDS ordination, as manually replacing the genus names resulted in all three primer sets clustering together (Appendix A). Lachnospiraceae are considered to be part of the core microbiome of the human gut [68,69]—all are anaerobic, and some taxa may be involved in intestinal diseases [69]. Their identification in jellyfish tentacles, which are likely to be an oxygen abundance niche, was therefore surprising. In contrast, the Spiroplasmataceae family and its class Mollicutes (identified using BLAST from the ostensibly Lachnospiracea ASV sequences) have previously been found in other jellyfish such as *Rhizostoma pulmo*, *Cotylorhiza tuberculate,* and *Aurelia aurita* [53,70,71]. The biology of the Spiroplasma marine lineages is still poorly understood, although they were suggested in sea cucumbers and in some terrestrial invertebrates to have diverse defensive capabilities (e.g., production of toxins and immune system recruitment) [72,73]. The reason for the potential misclassification is unclear, but the low level of identity with the Spiroplasma sequences in the SILVA database (~82–89%) yet high identity with sequences from a different jellyfish identified using BLAST (99.5–100%) suggest that this is not a technical artifact due to the error rate of the PCR enzyme used. Rather, this misclassification may have occurred because the representation of some clades in the SILVA database is still limited. Regardless, the actual identity of the “Spiroplasma-like” organism is still unclear. More broadly, potential misclassification will be hard to identify using a single primer set—we looked more deeply into this clade only because of the observed differences between the three primer sets. We therefore recommend including a comparative analysis using several different primers on a small, representative subset of any large-scale amplicon-based microbiome study of a new model system.

### 4.3. V3V4 Primers Are Commonly Used but Not without Caveats

We included the V3V4 primers [28] in our comparison as they were adopted in the official Illumina protocol [74] and have been extensively used in a variety of niches from environmental studies to the human gut microbiome [75,76,77], as well as in studies of corals and jellyfish (Figure 1B, Appendix A). A detailed, cross-taxa comparison also recommends this primer set over the V1V2 one due to several reasons, among them its better performance in identifying mock community composition and its ability to yield better functional imputations [26].

Given the challenges described above with the other two primer sets, we chose to use these primers for our ongoing, large-scale analysis of jellyfish-associated bacterial communities. The V3V4 primers are not without their own caveats, however, chiefly their lower amplification efficiency for SAR11, the most abundant heterotrophic clade in large regions of the ocean [27,56]. In the case of our jellyfish samples, the V4V5 primers did not identify SAR11 as a dominant member of the jellyfish microbiome. We do not expect using these primers to result in missing an important member of the microbial community. Another drawback of the V3V4 primers is their inability to amplify archaea and micro-eukaryotes, which were not very common in the results of our V4V5 primers from the same jellyfish tissue.

We note that there are several other primer pairs commonly used for cnidarian microbiome analyses, including the V1V3, V4, and V5V6 primer sets (Figure 1). The V1V3 has been used in the human microbiome project [23], but we did not include it in our analysis as the V1V2 primers are used more in jellyfish studies. The V4 region is shorter compared with primers that span more than one region and can be sequenced using a highly cost-efficient platform (e.g., using 2 × 150 base paired-end reads). It has been endorsed by the Earth Microbiome Project, which is a large-scale survey of environmental microbial communities [22]. However, several studies have suggested that longer amplicons are preferable [78,79,80]. The V5V6 primers are considered a good fit for coral microbiome studies [81] but have not been frequently used in other cnidarian systems. Future analyses including comparisons of these primers may help define an “optimal” set for primers for host-associated microbes.

### 4.4. Can the Results of Different Primer Sets Be Compared?

Over the last two decades, there have been thousands of studies of microbial ecosystems performed using 16S rRNA amplicons, and much of these data are publicly available. For example, there are ~150 papers in which cnidarian microbiomes have been studied using 16S rRNA gene amplicon sequencing, published since 2015. It would potentially be highly fruitful to directly compare these studies, both in order to identify large-scale trends in microbial populations in space and time, and in order to allow direct comparison between related studies (e.g., different jellyfish species). However, as demonstrated in Figure 1B, these studies often utilize different PCR primers. At least for the specific biological question addressed here (jellyfish microbiomes across tissues), the community composition dynamics did not differ between the three primer sets. Notably, when the taxonomic classification of specific ASVs differed between primer sets, using multiple primer sets allowed for the identification and, potentially, correction of these discrepancies. Taken together, our results suggest that the overall trends observed by the three primers could be directly compared, and merging these data into a unified dataset is also possible, albeit at a lower phylogenetic resolution (genus rather than ASV). However, it is important to note that our observations from samples that were processed identically and in parallel (i.e., sample collection, DNA extraction, and sequencing) may not be fully relevant for inter-study comparisons, where many other stages of the analysis workflow may introduce biases (reviewed in Pollock et al., 2018; Abellan-Schneyder et al., 2021).

## 5. Conclusions

There are many studies discussing the advantages and disadvantages of different primers, with no consensus [61,80,82,83,84]. Therefore, it is likely that future studies will continue to employ multiple different primers, depending on the specific model system, the preferences of the research team, and the importance of comparability with other studies. In studies of *R. nomadica* (and potentially other marine host–bacteria systems), we recommend using the V3V4 primer set, due to the advantages described above, and also because results using this primer set may be easier to compare with other publications on cnidarians, especially corals (Figure 1B), as well as with some cross-taxa studies [26]. However, because it is difficult to identify any specific primer set that is universally better than others, we recommend performing two steps before initiating any large-scale study of a new model. Firstly, we recommend performing a preliminary literature review of the primers used in related studies or organisms. Secondly, we advise testing several primers on a limited number of samples to identify (and potentially correct) technical issues, including misclassifications of major members of the microbial community. These stages can also help identify cases where multiple datasets using different primers can be compared.

## Figures and Tables

**Figure 1 microorganisms-11-00955-f001:**
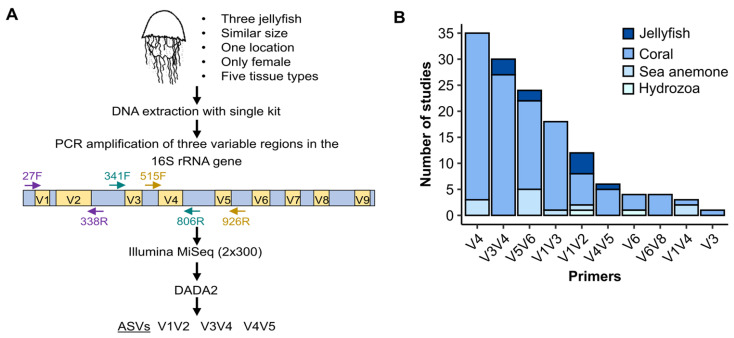
Overview of analysis stages used in this study. (**A**) Schematic illustration of the processing stages. DNA from jellyfish tissues was extracted. Amplicons were generated using different primer pairs targeting different variable regions and sequenced on an Illumina MiSeq. Afterwards, the sequences went through the same bioinformatic workflow on DADA2 and ASVs tables were created for each primer. (**B**) Number of studies and primer set types used for Cnidaria microbiome characterization since 2015.

**Figure 2 microorganisms-11-00955-f002:**
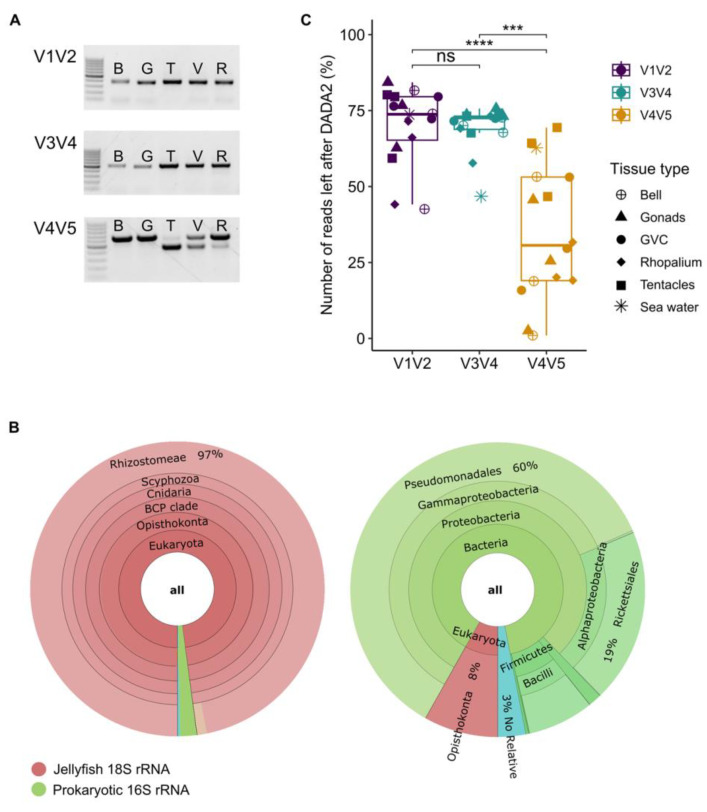
The double-band effect on downstream analysis. (**A**) PCR amplicons products for five jellyfish tissues, and three primer sets. B—bell, G—gonads, T—tentacles, V—gastrovascular canals, and R—rhoplium. (**B**) Percentage of initial reads remaining after DADA2 processing. The V4V5 remaining reads were significantly less than the other two primer sets (Kruskal–Wallis and Dunn’s tests. *p* < 0.001). (**C**) Visualization of relative abundance sequences in high vs. low bands obtained with V4V5 primers with the Krona RSF display tool [46]. *** and **** indicate *p*-value < 0.001 or 0.0001 level, respectively, while “ns” indicate non-significant results.

**Figure 3 microorganisms-11-00955-f003:**
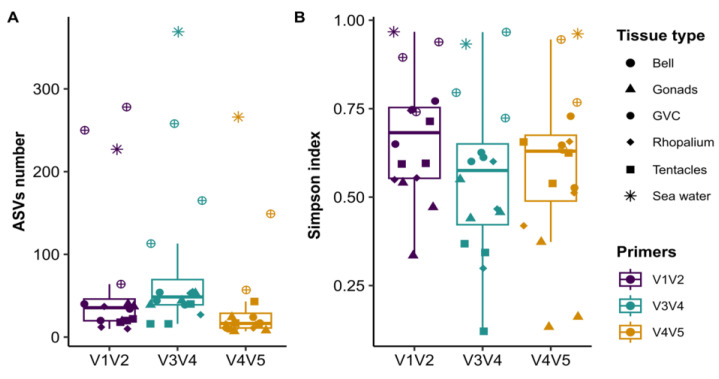
Alpha diversity indexes. (**A**) ASVs numbers. (**B**) Simpson’s Diversity Index. On both alpha diversity indexes, no significant differences were seen between primers (ANOVA, *p* > 0.05).

**Figure 4 microorganisms-11-00955-f004:**
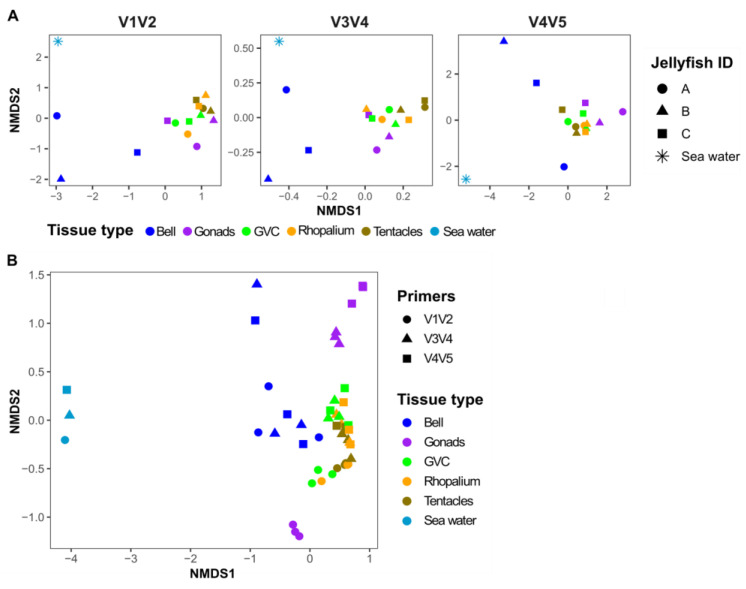
Non-metric multidimensional scaling (NMDS) of the bacterial community composition in different jellyfish and tissues obtained with different primers. (**A**) ASVs-based NMDS of the V1V2 primer (stress 0.07), V3V4 primer (stress 0.08), and V4V5 primer (stress 0.07). (**B**) Taxonomy-based NMDS analysis. Bacterial community composition of both ASVs and taxonomy-based NMDS are based on the Bray–Curtis dissimilarity (stress 0.10).

**Figure 5 microorganisms-11-00955-f005:**
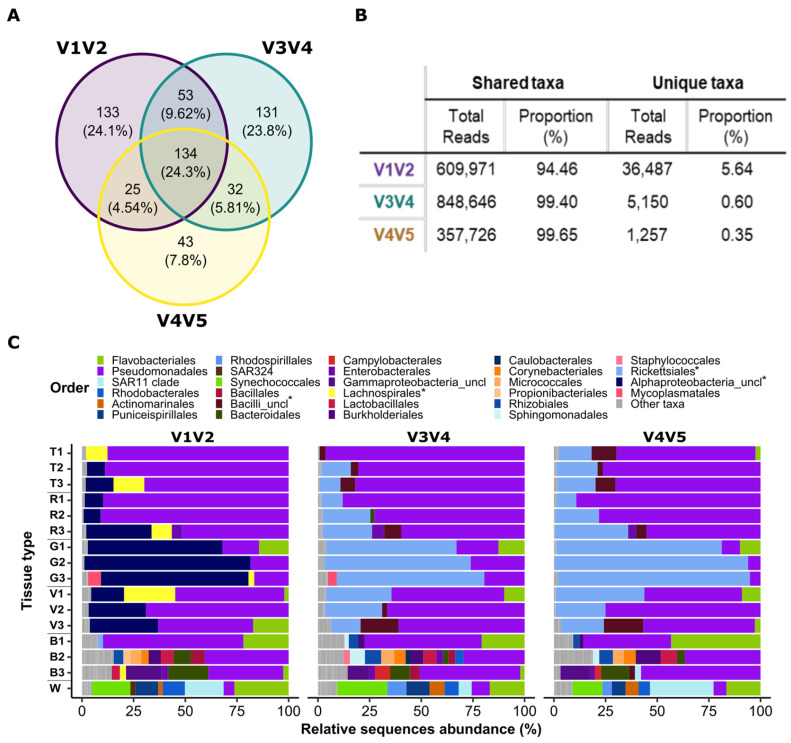
V1V2, V3V4, and V4V5 merged datasets analysis. (**A**) Venn diagram representing all genera from the merged datasets. (**B**) Reads numbers and proportion of shared and unique genera compared with the complete population of each primer. (**C**) Taxonomic compositions (relative sequence abundance) of order level in different tissues of 3 jellyfish (B—bell, G—gonads, T—tentacles, V—gastrovascular canals, and R—rhoplium) and one sample of filtered seawater (W). Taxa with relative sequence abundance of less than 2% were classified as “Other taxa“. * marks the main taxa discussed in this paper.

## Data Availability

The data that support the findings of this study are available in GitHub at https://github.com/NogaBarak/Selecting_16S_rRNA_primers_R.nomadica accessed on 8 March 2023. The raw sequencing data were deposited in the NCBI Sequence Read Archive (SRA) under accession number PRJNA870285.

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
