# Peer review of "Selecting 16S rRNA Primers for Microbiome Analysis in a Host–Microbe System: The Case of the Jellyfish Rhopilema nomadica"

_microorganisms, 2023, doi:10.3390/microorganisms11040955_

Round 1

Reviewer 1 Report

This article is really relevant. There is indeed a problem of choosing primers for 16S sequencing, as well as comparing the results obtained using different pairs of primers. Unfortunately, the number of samples tested does not allow us to confidently answer the questions posed by the authors. I understand that this work was carried out in order to select the primers for the authors' project and was not an end in itself, but it would be great if the authors would increase the number of jellyfish samples while reducing the number of tissues tested.
At the same time I would like to note that the methodical part of the work is well done.
You used the microbial community standard. The article lacks detailed results comparing V1V2, V3V4, and V4V5 for this control. Were the results for all the primers tested identical on this control?

There are also some typos in the article. For example, on line 347 "however".

Reviewer 2 Report

Dear authors,

In the present study “Selecting 16S rRNA primers for microbiome analysis in a host-microbe system: the case of the jellyfish Rhopilema nomadica” by Barak et al., the authors performed an extensive literature survey of studies focusing on amplicon sequencing approaches to describe the microbial composition of Cnidarian-associated microbiomes. The authors further conducted a comprehensive 16S amplicon sequencing approach with their model organism Rhopilema nomadica using three different primer sets spanning the mostly used hypervariable regions. All primer set revealed a similar pattern in bacterial community composition. However, the authors declared the performance of the V3V4 primer set as superior to V1V2 and V4V5. The authors propose that the V3V4 primer set is the most suitable for studying jellyfish-associated bacterial communities, although they recommend testing different primer sets for new model organisms.

 The manuscript addresses a recurring and long-standing discussion on the use of primers to sequence microbiomes. A particular focus is on host-associated microbomes, the analysis of which is particularly difficult due to contaminations with host components and host DNA. The authors advocate a careful selection of methods and finally primers that allow a comparison of different studies. The methods and results seem robust. Consequently, the authors recommend the use of V3V4 primer for the 16S analysis of at least jellyfish-associated microbiomes, particularly for their model jellyfish. I have two concerns to raise, which are explicitly commented in the attached pdf. First, the authors used a Taq polymerase for a two-step PCR approach without any proof-reading activity. For my understanding, particularly comprehensive studies should avoid PCR bias and some results might purely result from the choice of polymerase. Second, I recommend toning down the statement that “V3V4 primer set is the most suitable for studying jellyfish-associated bacterial communities”, since several studies and authors results show that the choice on primers depends on the model and even tissue.

 The abstract appropriately summarizes the study; however, I recommend toning down the above mentioned statement. The introduction covers all necessary points to understand the following results and methods. The methods part left behind a few specific questions, which should be answered in a point-by-point response. The results and discussion are robust, understandable and clearly presented with respective figures and table. 

Round 2

Reviewer 1 Report

Thank you to the authors for listening to my point of view. With corrections in the text and disclosure of the results of the control community sequencing, the article can be accepted for publication.